# Myoelectric and Inertial Data Fusion Through a Novel Attention-Based Spatiotemporal Feature Extraction for Transhumeral Prosthetic Control: An Offline Analysis

**DOI:** 10.3390/s25185920

**Published:** 2025-09-22

**Authors:** Andrea Tigrini, Alessandro Mengarelli, Ali H. Al-Timemy, Rami N. Khushaba, Rami Mobarak, Mara Scattolini, Gaith K. Sharba, Federica Verdini, Ennio Gambi, Laura Burattini

**Affiliations:** 1Department of Information Engineering, Università Politecnica delle Marche, 60131 Ancona, Italy; a.mengarelli@staff.univpm.it (A.M.); r.mobarak@pm.univpm.it (R.M.); m.scattolini@pm.univpm.it (M.S.); e.gambi@staff.univpm.it (E.G.); l.burattini@staff.univpm.it (L.B.); 2Biomedical Engineering Department, Al- Khwarizmi College of Engineering, University of Baghdad, Baghdad 10071, Iraq; ali.altimemy@kecbu.uobaghdad.edu.iq; 3Transport for NSW Alexandria, Haymarket, NSW 2008, Australia; rkhushab@gmail.com; 4College of Dentistry, Ibn Sina University of Medical and Pharmaceutical Sciences, Baghdad 10018, Iraq; g.kadhim@yahoo.com; 5Department of Theoretical and Applied Science, Università eCampus, 22060 Como, Italy; federica.verdini@uniecampus.it

**Keywords:** myoelectric control, pattern recognition, shoulder joint, spatiotemporal feature extraction, transhumeral amputees

## Abstract

This study proposes a feature extraction scheme that fuses accelerometric (ACC) and electromyographic (EMG) data to improve shoulder movement identification in individuals with transhumeral amputation, in whom the clinical need for intuitive control strategies enabling reliable activation of full-arm prostheses is underinvestigated. A novel spatiotemporal warping feature extraction architecture was employed to realize EMG and ACC information fusion at the feature level. EMG and ACC data were collected from six participants with intact limbs and four participants with transhumeral amputation using an NI USB-6009 device at 1000 Hz to support the proposed feature extraction scheme. For each participant, a leave-one-trial-out (LOTO) training and testing approach was used for developing pattern recognition models for both the intact-limb (IL) and amputee (AMP) groups. The analysis revealed that the introduction of ACC information has a positive impact when using windows of length (WLs) lower than 150 ms. A linear discriminant analysis (LDA) classifier was able to exceed the accuracy of 90% in each WL condition and for each group. Similar results were observed for an extreme learning machine (ELM), whereas k-nearest neighbors (*k*NN) and an autonomous learning multi-model classifier showed a mean accuracy of less than 87% for both IL and AMP groups at different WLs, guaranteeing applicability over a large set of shallow pattern-recognition models that can be used in real scenarios. The present work lays the groundwork for future studies involving real-time validation of the proposed methodology on a larger population, acknowledging the current limitation of offline analysis.

## 1. Introduction

Bionic replacement of the entire upper limb is recognized worldwide as a key research topic that involves different aspects, i.e., mechanics, bioelectronic sensor design, and information processing [1,2]. Moreover, recent literature has emphasized the critical role of sensory feedback, such as vibrotactile or visual, to restore proprioception, enhancing embodiment and enabling more natural interaction with the environment [3]. The first distinction that has to be faced in designing prosthetic solutions is related to the level of amputation. The large variety of causes that can lead to limb severing can be categorized into seven levels of limb absences, i.e., transcarpal, wrist disarticulation, transradial, elbow disarticulation, transhumeral, shoulder disarticulation, and forequarter amputations [4]. Among them, transradial and transhumeral amputations are the most frequent [4]. The former generally involves the design of forearm sockets with a prosthetic actuated hand [5,6]. Such control schemes exploit the myoelectric information recorded from residual muscle activity of the forearm and transduced into electrical signals by means of surface electromyography (EMG), magnetomyography, or mechanomyography [5,6,7,8,9]. Then, this information is decoded by pattern recognition models that are tuned during training sessions.

Transhumeral amputations impose additional challenges in the design and control of prosthetic devices [10,11]. Indeed, this kind of amputation requires dealing with two joints, i.e., the elbow and wrist joints, along with an actuated hand that provides prehensile functions [7,11]. Moreover, the total absence of the forearm restricts the available myoelectric sources to the shoulder or chest muscles [2,12]. A typical solution to amplify the neural commands involves surgical procedures such as targeted muscle reinnervation [12], which consists of transferring the residual amputated nerves to nonfunctional muscles that act as neural amplifiers of motor commands [12,13]. In this way, neural activity related to forearm and hand motion is magnified, and patterns can be extracted in order to implement efficient pattern recognition-based control schemes [13].

It should be noted that targeted muscle reinnervation requires a complex surgical procedure that can result in prolonged rehabilitation [14]. For this reason, alternative solutions have been investigated to design full-limb prosthesis control schemes for transhumeral amputations by exploiting myoelectric information recorded from the shoulder and chest muscles [14,15,16]. In refs. [14,15], a prosthesis control solution was proposed based on the concept of compensation cancellation, resulting in a physics-based paradigm that can be used as a viable alternative to more general myoelectric pattern recognition control schemes [14,15]. However, such machine learning models can be sensitive to changes in EMG signals due to electrode shift, which imposes experimental sessions for re-tuning the control architecture [17]. Hence, in addition to EMG data, the introduction of kinematic or kinetic data has been proposed in order to achieve more stable classification performance [2,18].

In ref. [18], myoelectric pattern recognition for robotic-aided rehabilitation was improved by introducing features from torque information, whereas in ref. [2], accelerometric data (ACC) were used to enlarge the feature space with low-frequency content information, useful for characterizing the shoulder girdle for transhumeral prosthetic control applications. The introduction of ACC information in a pattern recognition scheme was also leveraged in hand gesture recognition applications, as demonstrated in [19]. Furthermore, in the aforementioned studies, feature extraction from EMG and ACC signals was treated separately for each sensor, thus eventually implementing information fusion at the decision level [2]. Instead, it has been shown that including information fusion methods directly within the feature extraction scheme can provide significant advantages in terms of classification performance for myoelectric pattern recognition [20,21,22]. However, no previous work has explored the possibility of applying feature extraction procedures driven by information fusion between EMG and ACC channels for myoelectric pattern recognition on individuals with transhumeral amputation. Moreover, both EMG and ACC features have been computed on sliding windows greater than 150 ms [2,18,19]. However, as shown in ref. [16,23], good pattern recognition performance can also be obtained with a window length (WL) lower than 150 ms using a sparse EMG setup. In this context, the detrimental effects of reducing WL were observed at 50 ms, showing a reduction in classification performance in both individuals with intact limbs (IL) and those with transhumeral amputation (AMP).

The purpose of this work was to investigate the additional value provided by inertial information for the identification of shoulder movements in patients who underwent transhumeral amputation. Contributions to the existing literature can be summarized as follows. First, a novel information fusion scheme at the feature level was proposed to combine EMG and ACC data through an attention-based spatiotemporal warping architecture (A-STW) [21]. In this way, information coming from different sensors located on the chest and shoulder, i.e., EMG and ACC probes, was combined to assess whether the extracted features have better capabilities in separating shoulder motion commands in comparison with the use of EMG data only. Furthermore, the abovementioned fusion scheme was applied to the shoulder girdle of patients with transhumeral amputation. Finally, it was hypothesized that A-STW fusion would allow a reduction in the detrimental effects due to the reduction in the window length (WL) used for extracting handcrafted features [16,23]. To verify this, different WLs were used for feature extraction, down to 50 ms.

## 2. Materials and Methods

### 2.1. Experimental Protocol and Data Preprocessing

A dataset of 6 participants with intact limbs (IL) and 4 with transhumeral amputation (AMP), collected in ref. [24], was used for this study. Before the beginning of the first trial, participants were instructed regarding the experimental protocol, and they gave their written consent. Each experiment was carried out in accordance with the Declaration of Helsinki [24]. Demographic details of the 4 AMP participants are reported in Table 1. All participants with amputation were male and presented with a transhumeral level of amputation, which is less commonly represented in the literature than more distal cases [4]. Information regarding age, sex, amputated side, and time since amputation was collected to provide additional context, given the limited availability of such data in the literature [24].

For each participant, a total of 5 differential EMG channels were placed respectively on the pectoralis minor, the serratus anterior, the rhomboid major, and the lower and upper fibers of the trapezius. Electrodes were placed following the guidelines proposed in ref. [25]. Furthermore, a 3-axis accelerometer (ACC) ADXL335 (Analog Devices, Inc., Wilmington, MA, USA) was placed on the shoulder to identify the time points at which the shoulder started and stopped moving [2,24]. This sensor setup was designed to maximize spatial coverage while remaining comfortable for participants and suitable for potential integration into a smart prosthetic socket [10,16]. Additional chest muscles were avoided to prevent the setup from becoming cumbersome, while the ACC sensor near the deltoid ensured complementary information to the trapezius EMG. Both the accelerometer and EMG signals were acquired synchronously at a sampling frequency of 1000 Hz using an NI USB-6009 (National Instruments, Austin, TX, USA) (see Figure 1) [16,24]. During the experiment, participants were asked to provide feedback regarding their experience with the recording setup. None of them reported any discomfort either during the trials or during the sensor placement by the qualified personnel.

Participants were asked to perform 8 trials, each including 6 consecutive shoulder movements, i.e., elevation, depression, protraction, retraction, upward rotation, and downward rotation. A resting period between two consecutive movements was considered to prevent fatigue [2,16,24]. Then, for each participant and each trial, a signal preprocessing pipeline was applied for noise removal and signal epoch identification. In particular, a 4*th*-order, zero-phase Butterworth digital filter was applied to the EMG signals, with a passing band of 30–450 Hz, to remove components due to the electrical activity of the heart [2,16]. Accelerometer data were band-pass filtered in the range of 2–20 Hz [16]. A total of seven classes of movements were considered, i.e., the six shoulder movements and the rest condition, which was included in the pattern recognition problem, as suggested by [16,26,27].

### 2.2. A-STW Feature Extraction Architecture

Based on ref. [21], a novel spatiotemporal warping feature extraction architecture was developed by introducing an attention-based mechanism (Figure 2). This scheme takes advantage of the sparsity of the setup by capturing synergistic effects among muscles with respect to conventional approaches that extract features separately from each signal, eventually working only in time [20,21].

The proposed architecture is composed of two main blocks, which account for the spatial and temporal relations between different data. The spatial information was extracted by the dynamic time warping (DTW) between each pair of signals in the input, eventually resulting in a similarity matrix [21]. In more detail, consider two temporal sequences of length *l*: $s=\left(s_{1},s_{2},\ldots ,s_{l}\right)$ and $u=\left(u_{1},u_{2},\ldots ,u_{l}\right)$, which represent two signals. One can define D(s,u) as an l×l distance matrix between s and u, with each matrix element computed as $Dij={\left(s_{i}-u_{j}\right)}^{2}$. While the Euclidean distance is commonly employed by many approaches, it suffers from two main drawbacks: it is impractical for sequences of unequal length, and it is sensitive to time shifting. These issues can be addressed by finding the optimal path using DTW [21].

When using DTW, the distance is computed by following a specific warping path P, which is generated by traversing the matrix D along pairs of ordered positions, i.e., P=<$\left(e_{1},f_{1}\right),\left(e_{2},f_{2}\right),\ldots ,\left(e_{l},f_{l}\right)>$, where, $e_{i}\in \left[1:l\right]$ and $f_{i}\in \left[1:l\right]$ represent the positions of the warping path. To be considered a valid warping path, it must meet the conditions $\left(e_{1},f_{1}\right)=\left(1,1\right)$, $\left(e_{l},f_{l}\right)=\left(l,l\right)$, $0\leq e_{i+1}-e_{i}\leq 1$, and $0\leq f_{i+1}-f_{i}\leq 1$ for ∀i<l.

In general, a warping limit is imposed on the distances, indicated by $\left|e_{i}-f_{i}\right|\leq w\cdot l$, for all $\left(e_{i},f_{i}\right)\in P^{*}$, with *w* representing the maximum allowed deviation of the warping path from the diagonal. Therefore, the distance *D* along any path P is computed as(1)$$D_{P}\left(s,u\right)=\sum_{i=1}^{k}p_{i}$$
where $p_{i}=D_{ei,f_{i}}$ denotes the distance between the element at position $e_{i}$ of s and the element at position $f_{i}$ of u for the $i^{\mathrm{th}}$ pair of points in the proposed warping path P. Hence, considering the space of all possible paths as P, the DTW path $P^{*}$ is the one that yields the minimum distance:(2)$$P^{*}=\min_{P\in P}D_{P}\left(s,u\right)$$

The spatiotemporal warping feature extraction is pursued through the temporal information fusion, as highlighted in Figure 2. Here, the feature vector extracted from the current set of windows at time *t*, denoted as $DTW_{t}$, is added to a scaled version (scaling factor β) of the cell state component (c). The architecture exploits the concept of the cell state to account for long-term memory information [28,29]. This process can be understood as an information highway that traverses through all cells across multiple time steps, leading to a recursive representation of the features. A parameter β is employed to control the cell information update, as follows:(3)$$f_{t}=DTW_{t}+\beta c$$
In this context, the variable $f_{t}$ represents the extracted features at the current time *t*, and *c* denotes the cell state that is regulated by β, which was set to 0.75 [21]. Given this architecture, a cross-attention mechanism was introduced to realize the A-STW feature extraction scheme, which acts as an information fusion step, where attention masks from the features of the previous window are used to highlight the important features extracted from the current set of windows. This is then added to the cell state and used to update the cell in accordance with the scheme reported in Figure 2. It deserves to be noticed that the entire feature extraction scheme, although inspired by deep learning methodologies, is fully data-driven and does not require any learning processing. The attention mechanism between $DTW_{t}$ and $DTW_{t-1}$ can be mathematically expressed by:(4)$$\begin{array}{c} \mathrm{Softmax}\left(\begin{array}{c} \frac{DTW_{t}\cdot DTW_{t-1}^{T}}{\sqrt{d_{k}}} \end{array}\right)\cdot DTW_{t} \end{array}$$
where ${\left(\sqrt{d_{k}}\right)}^{-1}$ is a scaling factor [21,30]. It should be noted that the introduction of the attention mechanism with the softmax function represents a modification of the previous time-fusion mechanism presented in the literature, and it also guaranteed the mitigation of outliers in the feature space, which can naturally arise when dealing with EMG handcrafted feature extraction.

Thus, the A-STW architecture was fed according to [21,28,29], but different from previous works, here, EMG and ACC data enter at this level, contributing to spatial fusion. Hence, for each channel, the signal itself and the first and second discrete differentiation are included. This eventually leads to feature spaces with dimension given by $3\times (N_{c}\times \left(N_{c}-1\right))/2$, where $N_{c}$ is the number of channels.

To verify the hypothesis that the introduction of ACC data can produce more distinctive features, two conditions were investigated: feeding A-STW with only EMG channels (EMG condition) and with EMG and ACC data (EMG+ACC condition). Furthermore, these aforementioned conditions were investigated by considering WL epochs of the signals at 50, 100, and 150 ms. This allowed testing of the hypothesis in a range of processing window lengths suitable for myoelectric control [16,23], and it allowed the assessment of whether the proposed feature extraction scheme can provide more stable classification performances for a short WL compared to using EMG alone.

### 2.3. Clustering Properties Evaluation

The separability index (SI) was used to quantify the distance between classes in a given feature space [31]. This index was calculated for each participant and each trial in the two conditions mentioned in Section 2.2. In particular, given the covariance matrix of class *j*, denoted as $\Sigma_{j}$, and the covariance matrix of its most conflicting class, $\Sigma_{Cj}$, one can calculate the covariance matrix Σ as follows:(5)$$\Sigma =\frac{\Sigma_{j}+\Sigma_{Cj}}{2}$$
Next, the SI (separability index) is computed using the following formula:(6)$$\mathrm{SI}=\frac{1}{K}\sum_{j=1}^{K}(\frac{1}{2}\{{{\left(m_{j}-m_{Cj}\right)}^{T}\Sigma^{-1}\left(m_{j}-m_{Cj}\right)}^{\frac{1}{2}}\})$$
where *K* represents the number of clusters, $m_{j}$ denotes the mean of the data points in the jth cluster, and $m_{Cj}$ is the mean of the data points in the most conflicting cluster with respect to the jth cluster [31]. It is worth noting that the SI metric reflects the distances between classes in the feature space. Hence, a higher SI value indicates a better mapping of different movements in the given EMG feature space.

Pairwise comparisons between the SI obtained in the two conditions and for each WL were assessed using the Wilcoxon rank-sum test (α=0.05).

### 2.4. Pattern Recognition Models

In this study, a total of 4 traditional classifiers were used: linear discriminant analysis (LDA), *k*-nearest neighbors (*k*NN), and extreme learning machine (ELM) [9,32]. Moreover, a classifier proposed in refs. [33,34], i.e., the autonomous learning multi-model classifier of order zero (ALMMO), was considered due to its characteristics of being non-parametric, non-iterative, and fully autonomous.

For each participant (AMP and IL groups), a leave-one-trial-out (LOTO) training and testing approach was used to develop pattern recognition models. More specifically, the data from each trial underwent preprocessing and feature extraction as described in Section 2.1 and Section 2.2, respectively. Then, one trial at a time was left for testing while the remaining data were used for training. The final results are provided as the average of all the trials when used for testing. Performances obtained with EMG and EMG+ACC conditions were compared using the Wilcoxon rank-sum test (α=0.05).

### 2.5. Ablation Study: A-STW Multi-Modal Feature Extraction vs. Multi-Modal Feature Concatenation

To assess the effectiveness of the proposed A-STW approach, which, as shown in the spatiotemporal scheme illustrated in Figure 2, combines EMG and ACC signals through a spatiotemporal fusion architecture, an additional pattern recognition experiment was conducted using the best-performing classification model, i.e., LDA. Specifically, a LOTO training–testing scheme was applied to compare the EMG+ACC condition under A-STW with the *Merged* condition. In the *Merged* condition, the A-STW feature extraction scheme is applied separately to the EMG and ACC signals to extract distinct features, which are then concatenated before being fed to the classifier during training and testing within the LOTO framework. The test was performed for WLs of 150, 100, and 50 ms.

## 3. Results

### 3.1. Class Separability Properties

Mean SIs are reported in Figure 3, where values higher than 2.8 and 2.5 were observed for the IL and AMP, respectively, in the EMG+ACC condition for the three WLs employed. On the other hand, for the EMG condition, the SI showed significantly lower mean values, i.e., not greater than 2.3 and 2.1 for the EMG+ACC and EMG conditions, respectively. No significant differences were detected between different WLs within the same condition (EMG and EMG+ACC).

### 3.2. Classification Performances

Concerning the mean accuracy obtained in the LOTO validation scheme, one can observe mean accuracy values greater than 90% for both the IL and AMP groups (Figure 4a,b) using LDA and ELM with a WL of 150 ms in the case of EMG+ACC data, whereas *k*-NN and ALMMO show lower performances, i.e., an accuracy not greater than 87%. A slight but significant drop in performance was observed with a WL of 150 ms for the IL group in all the classifiers when using only EMG. The same can be stated for the AMP group, with the exception of ELM (Figure 4b), where the drop was not statistically significant.

In a similar fashion, a WL of 100 ms showed the same trend as previously reported for both the IL and AMP groups (Figure 4c,d). LDA again showed the best performance, with low standard deviation among participants in IL and AMP groups. Also, in this case, a drop in accuracy between EMG+ACC and EMG was observed for all the classifiers. However, it should be highlighted that in the case of EMG+ACC, an accuracy greater than 90% was obtained for both groups. The same holds when the data are processed with a WL of 50 ms (Figure 4a,f). However, in this case, the highest accuracy decreases were observed when passing from EMG+ACC to EMG, with the lowest classification performances observed for *k*-NN and ALMMO, i.e., mean accuracies lower than 81% and 78% for IL and not superior to 75% for AMP.

The analysis of the effects of decreasing WL is reported in Figure 5. It can be observed that, for both the IL and AMP groups, the EMG-only condition exhibited detrimental effects on classification performance, with a reduction in the mean accuracy and a higher variance across participants. Conversely, the EMG+ACC condition maintained stable performance, showing no consistent increase in variance and no noticeable drop in accuracy for either group.

### 3.3. Ablation Study: A-STW Multi-Modal Feature Extraction vs. Multi-Modal Feature Concatenation

An ablation study was conducted to compare the proposed integrated A-STW fusion (EMG+ACC) against a multi-modal feature concatenation approach (*Merged*) using LDA due to its superior performance. The results in Figure 6a demonstrate that A-STW fusion significantly outperformed the *Merged* condition in the IL group with a WL of 50 ms (*p* < 0.05) and maintained consistently higher classification accuracy with WLs of 100 and 150 ms. The superiority of A-STW is more emphasized in its performance in the AMP group (see Figure 6b), where the proposed approach consistently outperformed the *Merged* condition (*p* < 0.01 for a WL of 50 ms and *p* < 0.05 for WLs of 100 and 150 ms). This proves that the performance gain is specifically attributed to the power of the A-STW architecture in extracting meaningful neuromechanical features combining EMG and ACC at the data level, rather than merely the concatenation of features extracted independently from each modality.

## 4. Discussion

The study proposes a new feature extraction scheme that is able to fuse ACC and EMG at the feature level to provide better recognition of shoulder movements, which represents an important step toward the realization of prosthetic and assistive devices in patients with transhumeral amputation [16,35]. It should be noted that the feature extraction method proposed here is markedly different from that of many previous studies, in which features were extracted from EMG and ACC signals separately and information fusion appeared at the decision level [2,19,36]. On the contrary, here, fusion is realized through the A-STW approach at the signal level, as the distance between EMG and ACC also contributed to the numerical calculation of features. Hence, this further represents an element of the novelty of the study since the idea of using the presented feature extraction architecture to perform sensor fusion has not been investigated in previous similar works [2,16,19,36].

The results presented in Section 3.1 highlight that EMG+ACC guarantees significantly higher SIs compared with the EMG condition for both the IL and AMP groups (see Figure 3a,b). Moreover, the better separability properties of EMG+ACC held for all three WLs analyzed, meaning that the introduction of ACC information has a positive impact for WLs lower than 150 ms. This outlines that the proposed methodology has the potential to be applied in practical contexts, where short WLs are beneficial for developing real-time prosthetic control. In passing, many recent studies still computed features over temporal epochs equal to or even larger than 150 ms [2,19,36], indicating that dealing with short WLs (<100 ms) for feature extraction represents an issue that is far from being fully addressed [2,19,36]. It should be noted that the IL and AMP groups share a similar range of SI values, consistent with respect to the EMG+ACC and EMG conditions, indicating that A-STW produced separable clusters also when applied to impaired participants. A possible drawback is that the boost observed in class separability is obtained at the cost of enlarging the dimension of the feature spaces. Indeed, the dimension given by EMG+ACC was 84, whereas it was 30 for EMG only. However, the complexity induced by the introduction of ACC data can be handled by the modern microcontrollers for real-time applications, eventually mitigating the increase in computational burden carried by the A-STW sensor fusion approach.

Although SI represents a first assessment of the main clustering properties of a given set of features, it should be clarified here that it does not predict the eventual performance of pattern recognition architectures [31]. Thus, pairwise comparisons between classifiers using EMG+ACC and EMG modalities were conducted for both the IL and AMP groups to quantify, through classification accuracy, the added value obtained by the increase in class separability. As shown by the results (Figure 4), a significant boost in the classification performance for the IL group was observed in all the classifiers and the three WLs analyzed. The same outcomes were observed for the AMP group, with the exception of the ELM classifier, where significant improvements were provided by EMG+ACC only for 50 ms WL. It should be noted that LDA was able to achieve an accuracy of 90% in each WL condition and for each group, confirming it to be a reliable classifier when used in different WL conditions and with different sensing technologies, i.e., sparse or dense EMG setups, acoustic mechanomyography [9], and when EMG and ACC data are combined [2,9,37]. For the same classification tasks, it can be observed that the present results outperform the ones obtained in ref. [2], where the best mean accuracy of 88% was observed among the participants for a WL of 500 ms, supporting the hypothesis that A-STW offers a viable alternative with respect to information fusion at the decision level. Furthermore, unlike previous studies, in which supervised feature reduction techniques like spectral regression were applied [37,38], no feature reduction was performed in this research. This was specifically done to investigate the direct application of A-STW.

The analysis further pointed out that the introduction of ACC data in the feature fusion scheme mitigated the detrimental effects generally caused by WL reduction [39]. Indeed, the EMG+ACC condition showed stable performance for AMP participants when the WL was reduced from 150 to 50 ms compared with EMG only, where a reduction in the mean accuracy was recognized for each WL (Figure 5), which lays the foundation for possible optimization procedures tailored to individual users. This may be attributed to the fact that, during the maintenance condition of a shoulder pose, the ACC data show stationary components due to the projection of the gravity acceleration along the three axes of the sensor [40,41]. In this case, such constant trends likely provide more clustered data points, as confirmed by the greater SI values (Figure 3). One may wonder whether the leading information that carries the best performance is due to the ACC signals. However, as shown in previous work [2], the ACC information alone did not provide good results for the specific task considered in this study. On the contrary, the findings of this study suggest that inertial signals have value for classification performance, as they permit the stabilization of the quality of the fused features. The latter aspect can have a strong impact on the quality of the machine learning model developed, as a reduced number of outliers is expected, and the A-STW appears to have beneficial effects on this kind of aspect. This observation is further supported by the jitter plot (Figure 7), where the feature distribution obtained with A-STW is compared to that of the STW alone. The comparison highlights a reduced number of outlier data points when the attention-based fusion of ACC and EMG signals is employed during feature extraction. This result confirms that the attention mechanism effectively stabilizes the feature space, mitigating the misclassification bursts observed with STW alone. In this sense, the analysis implicitly serves as an ablation-like evaluation at the feature level, demonstrating the crucial contribution of the attention mechanism to the improved and more consistent performance of the proposed A-STW approach.

The beneficial effects of fusing the two sources of information are supported by the latency requirement of 150 ms for real-time applicability [39]. Indeed, an offline evaluation of the LDA model showed that the latency introduced during testing, both for feature computation and classification, averaged only 3 ms when executed on an ASUS workstation with an Intel i5 processor. This is of great importance for the applicability of the framework in a real context, where unwanted movements of the prosthetic device should be avoided through opportune control mechanisms [42]. This is illustrated in Figure 8, which shows the mean confusion matrix among participants with amputation. The improvement is distributed across all classes, leading to more accurate class recognition and, ultimately, cleaner pattern recognition. To further confirm the effectiveness of A-STW in extracting neuromechanical features combining EMG and ACC, rather than concatenating independent features extracted separately from EMG and ACC, an ablation study, as described in Section 2.5, was performed using the LDA model under three different WL conditions (150, 100, and 50 ms). As shown in Figure 6, the EMG+ACC condition consistently outperformed the *Merged* condition across all WL values in the AMP group (Figure 6b), with the most pronounced improvement observed at 50 ms. This indicates that extracting EMG and ACC features separately and then merging them is not an optimal strategy, particularly for short WLs (i.e., faster update times). Instead, fusing EMG and ACC at the feature extraction architecture level yields superior results compared with concatenating separately extracted EMG and ACC features, as performed in previous work [2]. A similar trend was observed in the IL group, where significant differences were also found at 50 ms. For WLs of 150 and 100 ms (Figure 6a), the IL group showed slight improvements under the EMG+ACC condition, although these were not statistically significant. This suggests that, in scenarios where EMG and ACC signals are not affected by pathological conditions, the observed improvement in classification performance compared with previous studies [2,35] can be attributed to the proposed A-STW scheme.

Finally, it is important to address some limitations in future studies. While the protocol aligns with established methods for collecting high-quality data, achieving real-world applicability requires fast and efficient training of the underlying architectures. In this context, a recent work proposed a promising direction through a self-calibrating random forest model that enables plug-and-play myoelectric control [43]. This type of self-adaptive framework could serve as a foundation for home-based calibration scenarios that require minimal user intervention and reduced setup complexity. Even though this aspect falls outside the scope of the present study, it is important to acknowledge that strategies like fusing feature extraction schemes must eventually be integrated with higher-level control policies to achieve real-life applicability, particularly for individuals with transhumeral amputation.

This brings attention to a second, central point: future research must address the gap between offline feature analysis and real-time implementation. The present study was limited to offline analysis, a well-established approach in the field [9,35,44], but extending this methodology to a larger cohort of patients under real-time conditions is a fundamental step toward strengthening the credibility of sensor-fused interfaces for prosthetic control.

## 5. Conclusions

This study supports the hypothesis that ACC data can improve shoulder movement characterization and the performance of pattern recognition-based control systems for transhumeral prostheses. Moreover, the proposed A-STW architecture showed promising results in terms of class separability and motion identification accuracy. The use of ACC data eased the recognition of shoulder movements and mitigated the effects of WL reduction, supporting the development of more robust pattern recognition-based prosthetic control systems. Further investigations are needed to evaluate the proposed approach in a larger and heterogeneous population, as well as in real-life settings.

## Figures and Tables

**Figure 1 sensors-25-05920-f001:**
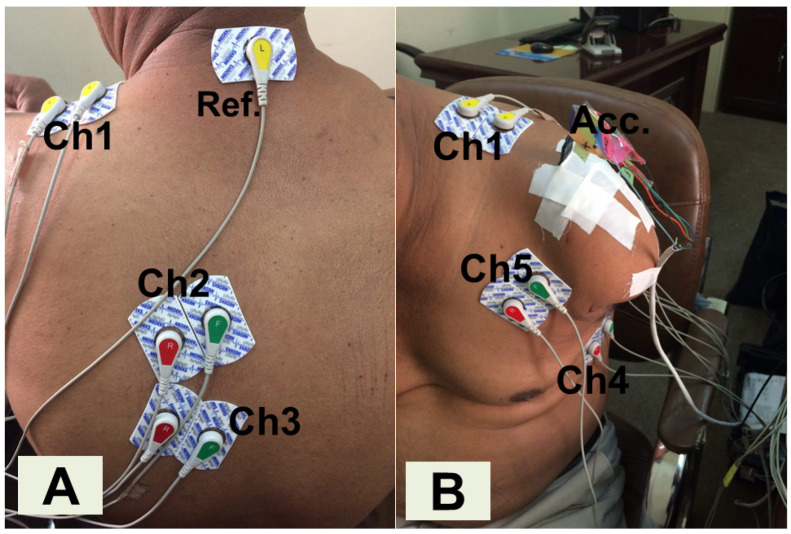
The experimental setup for recording EMG and ACC data in a transhumeral amputee. (**A**) shows the back of the patient, where Channel 1 (Ch1) and Ch3 recorded the myoelectric activity of the upper and lower fibers of the trapezius, whereas Ch2 was placed on the rhomboid major. The reference (Ref.) electrode was placed on the neck. (**B**) shows the front of the participant, the ACC sensor was placed on the shoulder, while Ch4 and Ch5 were placed on the serratus anterior and the pectoralis minor, respectively.

**Figure 2 sensors-25-05920-f002:**
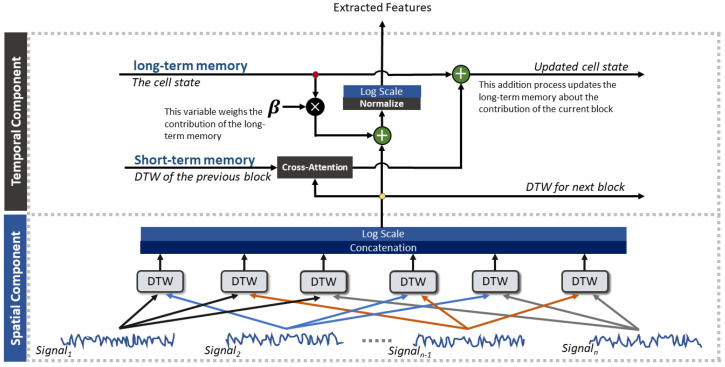
The spatiotemporal warping feature extraction method. A spatial component exploits DTW for fusing multisource data. Then, the temporal component uses short- and long-term memory to update the extracted features.

**Figure 3 sensors-25-05920-f003:**
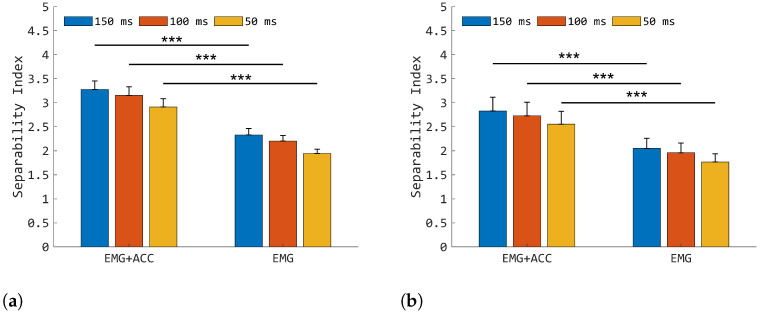
This figure shows the mean SI at 150, 100, and 50 ms for the IL (**a**) and AMP (**b**) groups. The symbol *** indicates *p* < 0.001.

**Figure 4 sensors-25-05920-f004:**
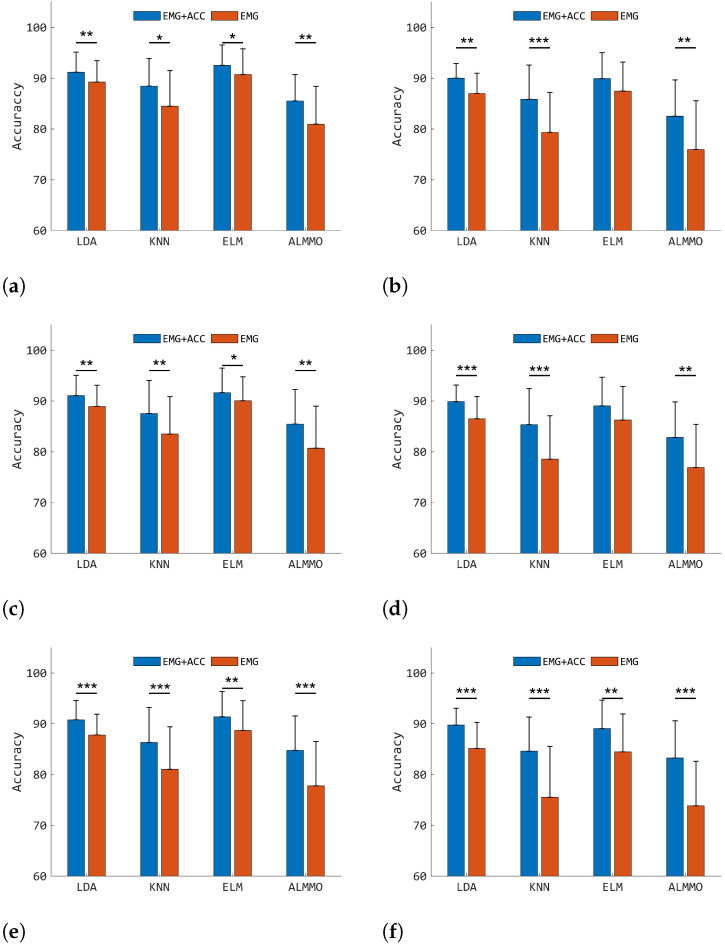
(**a**,**c**,**e**) show the mean accuracies in the IL group for WLs of 150, 100, and 50 ms, respectively. The blue bars refer to EMG+ACC conditions, while the red bars indicate the case of using only EMG data. In the same way, (**b**,**d**,**f**) show the mean accuracy for the AMP group in the same order of WLs and with the same representative colors. The symbols *, **, and *** indicate *p* < 0.05, *p* < 0.01, and *p* < 0.001, respectively.

**Figure 5 sensors-25-05920-f005:**
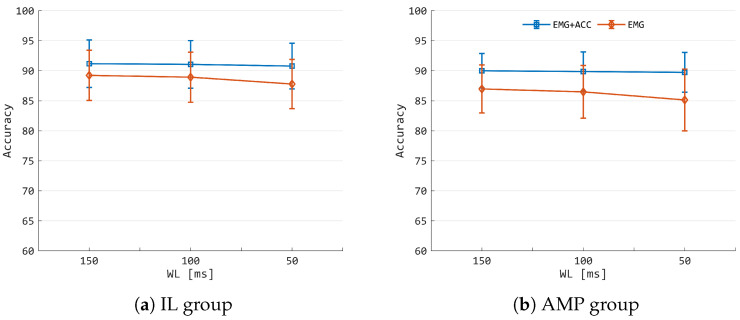
Mean classification accuracy of the LDA classifier across different window lengths (WLs). (**a**) refers to the IL group, and (**b**) refers to the AMP group. Blue lines indicate the EMG+ACC condition, while red lines indicate the EMG-only condition.

**Figure 6 sensors-25-05920-f006:**
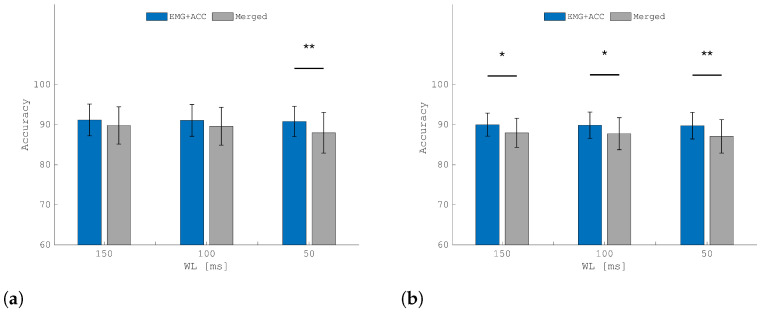
Results comparing LDA accuracy using A-STW with EMG+ACC and multimodal feature fusion (*Merged*) with WLs of 150, 100, and 50 ms for both the IL (**a**) and AMP (**b**) groups. The symbol * indicates p< 0.05, and ** indicates p< 0.01.

**Figure 7 sensors-25-05920-f007:**
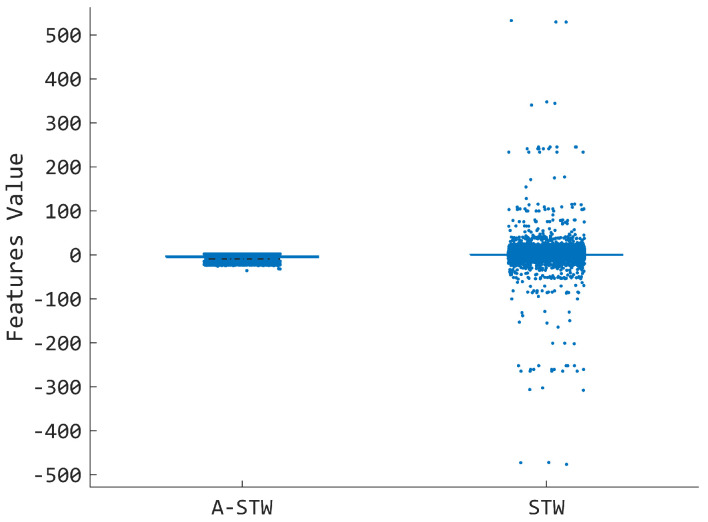
Jitter plot of the representative first component of the feature set generated for a participant with amputation when A-STW and STW were applied. Example reported for a WL of 150 ms.

**Figure 8 sensors-25-05920-f008:**
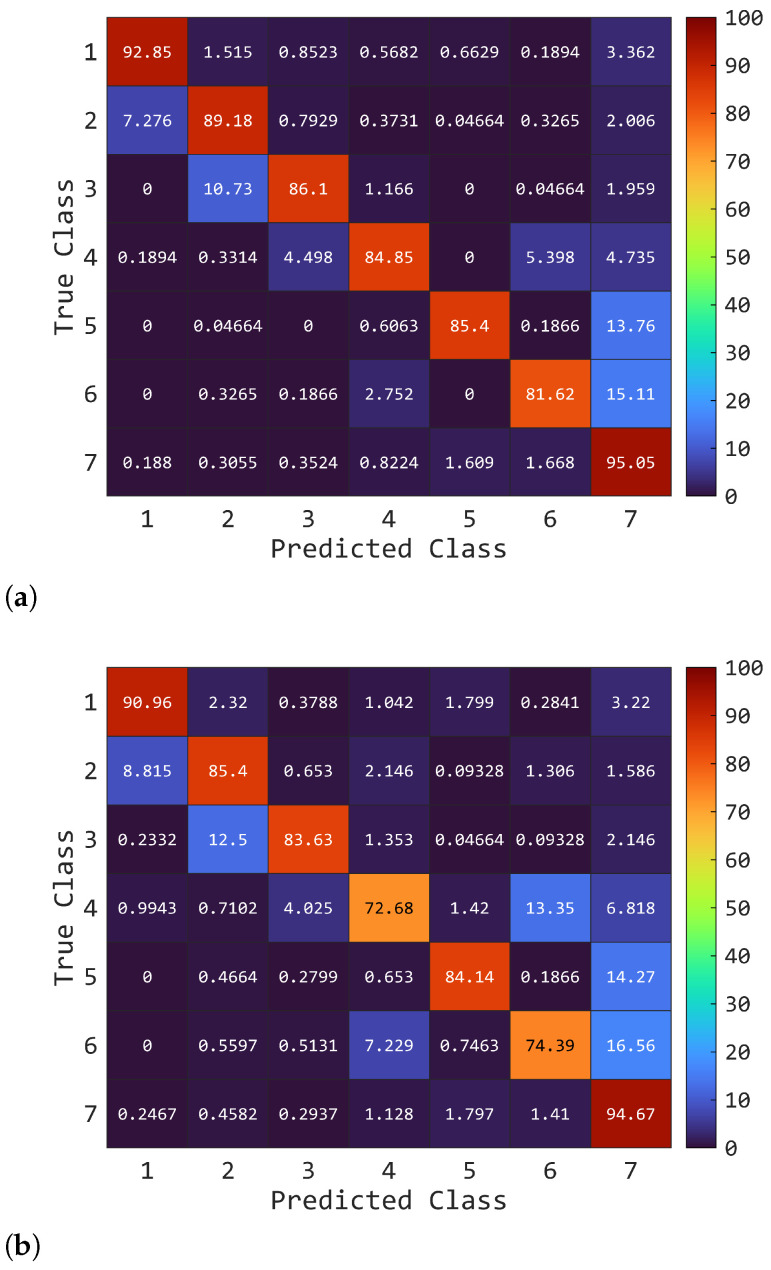
This figure shows the mean confusion charts among participants with amputation obtained during testing using LDA with features computed at 150 ms. (**a**) shows the case of using EMG+ACC, whereas in (**b**), the case of EMG only is reported. Classes represent the following: (1) elevation, (2) depression, (3) protraction, (4) retraction, (5–6) upward and downward rotation, and (7) rest.

**Table 1 sensors-25-05920-t001:** Demographic and clinical information of participants with transhumeral amputation.

Participant	Age (years)	Sex	Amputated Side	Time Since Amputation (years)	Cause of Loss of Arm	Wearing a Prosthetic Limb	Dominant Limb
A1	65	Male	Left	32	War	No	Right
A2	50	Male	Left	31	War	No	Right
A3	35	Male	Right	23	Terrorist bomb	No	Right
A4	16	Male	Right	11	Terrorist bomb	No	Right

## Data Availability

The dataset supporting the findings of this study is publicly available on the Kaggle platform under the title “Shoulder EMG Dataset for Amputees” (also known as the Transhumeral Amputees EMG Dataset). It includes electromyography (EMG) and accelerometer recordings of shoulder movements performed by participants with transhumeral amputation. Access is provided via Kaggle and is subject to the dataset’s usage terms and license. Researchers can download the data from https://www.kaggle.com/datasets/alihaltimemy/shoulder-emg-dataset-for-amputees.

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
