# Peer review of "Myoelectric and Inertial Data Fusion Through a Novel Attention-Based Spatiotemporal Feature Extraction for Transhumeral Prosthetic Control: An Offline Analysis"

_sensors, 2025, doi:10.3390/s25185920_

Round 1
Reviewer 1 Report
Comments and Suggestions for Authors
In this study, authors proposed to combine the EMG signals and ACC signals to improve the motion intention decoding performance for both intact-limb and amputees subjects. The authors mentioned some contributions; however, my major concern is that the results cannot provide sufficient evidences.
Firstly, unlike previous studies where the features are extracted from EMG and ACC signal separately, and information fusion is performed at decision level, this study proposed the A-STW architecture to perform the information fusion scheme at the feature extraction level. In order to verify its effectiveness, the control method should also utilize both EMG and ACC signals and do feature fusion at decision level. This study only compares the EMG+ACC and EMG, it is not surprising to achieve better classification results.
Secondly, the author claimed that the A-STW fusion would allow to reduce the detrimental effects of using a small window length. However, only Figure 5 compares the performance degradation between the two methods (EMG+ACC v.s. EMG) along the reduction of window length using LDA. The statistical analysis should be performed across conditions at the group level to further verify this. Again, the control method should also use EMG+ACC with feature fusion at the decision level.
Other aspects that can possibly improve the manuscript, including the major issues:
- Ablation studies need to be conducted to explicitly quantify the novelty of A-STW over STW.
- The computational cost of supporting real-time usage needs to be analyzed.
Minor issues:
- The figures can be further improved. For example, the font size is generally too small. There is no label for the y-axis of Figure 3(a). In Figure 4 and 5, although the caption indicates the meaning of different colors, it is recommended to illustrate the information using legend in the figures.
- It is recommended to put Figure 5 and 7 and related descriptions in the results section.
- Line 103, “to remove components due to the electrical activity of the hearth”, I guess it should be heart.
Author Response
We thank the Reviewer for the valuable comments. A detailed, point-by-point response has been provided in the attached file.

Reviewer 2 Report
Comments and Suggestions for Authors
Manuscript ID: sensors-3794018
Title: Myoelectric and Inertial Data Fusion Through a Novel Attention-Based Spatio-Temporal Feature Extraction for Transhumeral Prosthetic Control
This study addresses an important challenge in the control of prosthetic devices for transhumeral amputees. As transhumeral amputation involves the loss of the entire arm below the shoulder, the remaining musculature is limited, and voluntary motion is often restricted. Surface EMG signals alone typically provide limited information, making precise movement recognition difficult. Consequently, current commercial myoelectric prostheses are restricted to very basic control, lacking the ability to perform complex actions. By fusing accelerometric (ACC) and EMG data, this study takes a meaningful step toward enhancing control systems, and from a clinical perspective, it is a necessary direction for improving prosthetic usability in transhumeral amputees.
Nonetheless, I hope the manuscript can be further refined with the following suggestions to meet the publication standards of Sensors:
Abstract
- Background: While the research context and significance are explained, emphasizing the clinical and technical necessity would improve clarity.
- Method: Additional detail on the number of subjects, data acquisition equipment, and feature extraction methods would be helpful.
- Applicability: While potential applications are mentioned, it would be beneficial to include a clear statement of the study’s limitations and future directions.
2. Materials and Methods
- The study includes only a small number of participants (4 amputees and 6 intact-limb controls). Please provide demographic information such as age, sex, and time since amputation.
- EMG characteristics may vary depending on the level of amputation and rehabilitation status. It is important to present relevant medical/clinical data to enhance generalizability.
- Indicate dominant limb for each participant, as EMG signal characteristics may vary accordingly.
- The experimental protocol involved controlled, isolated shoulder motions in a fixed environment over short periods. This may not reflect real-life functional use. What strategies do you propose to bridge this gap?
- The current study relies on offline analysis. There is no mention of real-time control. Has real-time implementation been considered or tested?
- In real-time control, performance metrics such as latency, responsiveness, and robustness are critical. Please clarify if these were evaluated.
- For clinical relevance, testing with actual prosthesis use during daily activities (e.g., eating, grasping, walking with arm swing) would be valuable.
- The attention-based spatio-temporal warping model is described conceptually, but further clarification is needed regarding: Neural network architecture、 Number of parameters、 Training methodology、 Computation time
- From the user’s perspective, how were wearability, ease of use, learning time, and error rates evaluated?
- Including subjective user feedback would enrich the study.
- Was the integration of sensory feedback (e.g., vibrotactile, visual feedback) considered or discussed?
Additional Methodological Questions
- EMG electrodes were mainly placed on posterior trunk muscles. Given the shoulder’s multidirectional mobility (ball-and-socket joint), why were anterior and upper arm muscles (e.g., deltoid, biceps brachii) not included?
- Among the 7 classes (6 motions + rest), rest and similar low-movement classes (e.g., depression, retraction) may be difficult to distinguish. Was a confusion matrix or misclassification analysis presented?
- Preprocessing parameters (e.g., filtering, windowing) appear to be uniformly applied across participants. Given inter-subject EMG variability, were individual calibrations considered?
- Was MVIC (Maximum Voluntary Isometric Contraction) used for normalization?
3. Results & 4. Discussion
- The proposed EMG+ACC fusion-based system shows promising results for classifying shoulder movements in transhumeral amputees. The LDA and ELM classifiers demonstrated strong performance in both accuracy and computational efficiency, suggesting feasibility for real-time prosthetic control. However, the study would be greatly strengthened by addressing the methodological limitations outlined above.
References
- After reviewing the references, I found that seven papers from 2022 and two from 2023 were cited, but there are no citations from 2024 publications. Considering that the field of sensor technology is rapidly evolving, with many new studies being published, are there no more recent papers that could have been referenced?
I hope these suggestions help improve the manuscript and support its successful publication in Sensors.
Author Response
We thank the Reviewer for the valuable comments. A detailed, point-by-point response has been provided in the attached file

Round 2
Reviewer 1 Report
Comments and Suggestions for Authors
In Nsugbe (2022), the individual results presented for each subject show that EMG+ACC outperformed EMG alone under all conditions. Thus, despite the absence of statistical analysis, Nsugbe (2022) has provided reasonably sufficient evidence that fusing EMG and ACC information improves classification performance.
Compared to the results in Nsugbe (2022), a substantial improvement in classification performance is evident under both the EMG-only and EMG+ACC conditions. This enhancement likely stems from the newly proposed feature extraction network. The authors claim this network can fuse EMG and ACC information during the feature extraction stage, asserting this capability significantly boosts classification performance. I maintain my view that the authors should supplement their method with a comparative approach: separately feeding EMG and ACC signals into the feature extraction network, then merging the extracted features for classification. If the authors' method still surpasses this reviewer-suggested comparative approach, their conclusion would be undoubtedly valid. Otherwise, the observed improvement can only be attributed to the superiority of the feature extraction network over traditional time-frequency domain features, rather than any specialized ability of the multi-modal feature extraction network to fuse EMG and ACC information.
Author Response
Comment: In Nsugbe (2022), the individual results presented for each subject show that EMG+ACC outperformed EMG alone under all conditions. Thus, despite the absence of statistical analysis, Nsugbe (2022) has provided reasonably sufficient evidence that fusing EMG and ACC information improves classification performance.
Compared to the results in Nsugbe (2022), a substantial improvement in classification performance is evident under both the EMG-only and EMG+ACC conditions. This enhancement likely stems from the newly proposed feature extraction network. The authors claim this network can fuse EMG and ACC information during the feature extraction stage, asserting this capability significantly boosts classification performance. I maintain my view that the authors should supplement their method with a comparative approach: separately feeding EMG and ACC signals into the feature extraction network, then merging the extracted features for classification. If the authors' method still surpasses this reviewer-suggested comparative approach, their conclusion would be undoubtedly valid. Otherwise, the observed improvement can only be attributed to the superiority of the feature extraction network over traditional time-frequency domain features, rather than any specialized ability of the multi-modal feature extraction network to fuse EMG and ACC information.
Response: We thank the Reviewer for highlighting this point and for helping us improve the quality of the work. We acknowledge that the proposed feature extraction scheme can itself represent an improvement over previous literature and can provide superior results even when using EMG alone. For this reason, we performed the ablation study requested by the Reviewer for the LDA model with a 150 ms WL. Specifically, we retrained the model as suggested by the Reviewer, i.e., applying the feature extraction scheme separately to EMG and ACC, then merging the features before training and testing the LDA classifier, thereby implementing fusion at the decision level rather than at the feature extraction level.
For both the AMP and IL groups, the EMG+ACC condition outperformed the merged condition. The latter, however, showed intermediate performance between EMG alone and EMG+ACC. This outcome supports two conclusions: (i) the greatest improvement is obtained when the feature extraction architecture fuses EMG and ACC at the feature level; and (ii) the feature extraction scheme itself is highly effective, since we observed no consistent drop in accuracy and, in all conditions, our results exceeded those reported in previous studies.
In the revised Discussion section, we included the following part (highlighted in red):
To further confirm the effectiveness of fusing EMG and ACC information at the feature level rather than at the decision level, a specific study was performed for the LDA model with a WL of 150 ms. Indeed, an additional case, i.e., merged, was investigated, considering the use of A-STW for extracting features from EMG and ACC separately, then merging them and using the combined features to train the LDA classifier, thus implementing fusion at the decision level. The results are reported in Figure 8. It can be observed that, for both the AMP and IL groups, the EMG+ACC condition consistently outperforms the merged condition. This confirms that the greatest improvement is achieved by fusing EMG and ACC information at the feature level. However, it should be noted that part of the improvement can still be attributed to the STW-based scheme, since the use of EMG alone provides excellent results when compared with previous studies [2,16].

Figure 8. Results comparing LDA accuracy using A-STW with EMG, EMG+ACC and decision-level fusion (merged) with a WL of 150 ms, for both AMP and IL groups.
Reviewer 2 Report
Comments and Suggestions for Authors
Dear Authors,
This study tackles a significant challenge in controlling prosthetic devices for transhumeral amputees. Because transhumeral amputation results in the loss of the entire arm below the shoulder, the remaining musculature is limited, and voluntary motion is often restricted. Surface EMG signals alone generally provide insufficient information, making precise movement recognition challenging. As a result, existing commercial myoelectric prostheses are limited to basic control functions and cannot perform complex actions. By integrating accelerometric (ACC) and EMG data, this study takes an important step toward advancing control systems, and from a clinical standpoint, it represents a necessary direction for improving prosthetic usability for transhumeral amputees.
Thank you for your efforts in responding to multiple questions and revising the manuscript. I believe this research paper is suitable for publication in Sensors.
Thanks.
Author Response
We sincerely thank the Reviewer for the valuable feedback provided during the review process and for endorsing the publication following our responses to all comments.
Round 3
Reviewer 1 Report
Comments and Suggestions for Authors
The author has address all my comments and it can be accepted.